**www.cambridge.org/qrd**

# Structural bioinformatics studies of six human ABC transporters and their AlphaFold2-predicted water-soluble QTY variants

Emily Pan[1] ⓘ, Fei Tao[2] ⓘ, Eva Smorodina[3] ⓘ and Shuguang Zhang[4] ⓘ

[1]The Lawrenceville School, Lawrenceville, NJ, USA; [2]Laboratory of Food Microbial Technology, State Key Laboratory of Microbial Metabolism, School of Life Sciences and Biotechnology, Shanghai Jiao Tong University, Shanghai, China; [3]Laboratory for Computational and Systems Immunology, Department of Immunology, University of Oslo, Oslo University Hospital, Oslo, Norway and [4]Laboratory of Molecular Architecture, Media Lab, Massachusetts Institute of Technology, Cambridge, MA, USA

**Keywords:**
convert hydrophobic alpha helix to hydrophilic alpha helix; membrane protein design; protein structural predictions; QTY code; water-soluble integral membrane proteins

**Corresponding author:**
Shuguang Zhang;
Email: Shuguang@MIT.EDU

**Abstract**

Human ATP-binding cassette (ABC) transporters are one of the largest families of membrane proteins and perform diverse functions. Many of them are associated with multidrug resistance that often results in cancer treatment with poor outcomes. Here, we present the structural bioinformatics study of six human ABC membrane transporters with experimentally determined cryo-electron microscopy (CryoEM) structures including ABCB7, ABCC8, ABCD1, ABCD4, ABCG1, ABCG5, and their AlphaFold2-predicted water-soluble QTY variants. In the native structures, there are hydrophobic amino acids such as leucine (L), isoleucine (I), valine (V), and phenylalanine (F) in the transmembrane alpha helices. These hydrophobic amino acids are systematically replaced by hydrophilic amino acids glutamine (Q), threonine (T), and tyrosine (Y). Therefore, these QTY variants become water soluble. We also present the superposed structures of native ABC transporters and their water-soluble QTY variants. The superposed structures show remarkable similarity with root mean square deviations between 1.064 and 3.413 Å despite significant (41.90–54.33%) changes to the protein sequence of the transmembrane domains. We also show the differences in hydrophobicity patches between the native ABC transporters and their QTY variants. We explain the rationale behind why the QTY membrane protein variants become water soluble. Our structural bioinformatics studies provide insight into the differences between the hydrophobic helices and hydrophilic helices and will likely further stimulate designs of water-soluble multispan transmembrane proteins and other aggregated proteins. The water-soluble ABC transporters may be useful as soluble antigens to generate therapeutic monoclonal antibodies for combating multidrug resistance in clinics.

## Introduction

The ATP-binding cassette (ABC) transporter superfamily, characterized by their use of ATP hydrolysis to drive the transport of a variety of molecules across the plasma membrane and many intracellular membranes, is one of the largest transporter gene families. The biological functions and their relevance in diseases of the human transporters have recently been well studied (Theodoulou and Kerr, 2015; Dean et al., 2022). In humans, 49 subtypes have been identified and organized into seven subfamilies. Though typically the functional protein contains two nucleotide-binding domains and two transmembrane domains with 5–11 membrane spanning alpha helices, some are hemi-transporters and must form either homodimers or heterodimers to constitute a functional transporter (Dean et al., 2001). ABC transporters are also involved in many key cellular processes including i) maintenance of osmotic homeostasis, ii) antigen processing, iii) cell division, immunity, iv) cholesterol, and v) lipid trafficking (Liu, 2019). Expressed throughout the body, many subtypes are associated with the development of multidrug resistance as these proteins are involved in the a) absorption, b) distribution, and c) excretion of drugs (Liu, 2019). ABC transporters are also linked to various human diseases, including a) cystic fibrosis, b) tangier disease, c) immune deficiencies, sitosterolemia, d) Dubin–Johnson syndrome, e) pseudoxanthoma elasticum, and more (Table 1; Dean et al., 2001).

For example, ABCB7 is a member of MDR/TAP subfamily. It is found in mitochondria and plays a critical role in iron homeostasis, is linked to sideroblastic anemia with spinocerebellar ataxia, and is also involved in multidrug resistance and antigen representation for B-cell development and proliferation (Bekri et al., 2000; Taketani et al., 2003; Boultwood et al., 2008; Zutz et al., 2009; Lehrke et al., 2021; Yan et al., 2022).

ABCC8 has been shown to regulate the ATP-sensitive potassium channels including K(ir)6.2 and KCNJ11 (Bryan et al., 2007; De Franco et al., 2020). ABCC8 is found in pancreatic β cells and participates in glucose metabolism and is linked to various forms of hypoglycemia as well as neonatal diabetes mellitus (Haghverdizadeh et al., 2014).

**Table 1.** Characteristics of six human ABC native transporters in this structural bioinformatic study

| Name (Uniprot ID) | Structure (Å, PDB ID) | Tissue expression | Medical relevance |
|---|---|---|---|
| ABCB7 (O75027) | CryoEM (3.3 Å, 7VGF) | Mitochondria | Ataxia, anemia |
| ABCC8 (Q09428) | CryoEM (3.30 Å, 7S5V) | Pancreas | Sulfonylurea receptor |
| ABCD1 (P33897) | CryoEM (3.14 Å, 7SHM) | Peroxisomes | VLCFA transport regulation |
| ABCD4 (O14678) | CryoEM (3.60 Å, 6JBJ) | Lysosome | Neurodegenerative diseases |
| ABCG1 (P45844) | CryoEM (3.20 Å, 7R8D) | Ubiquitous | Cholesterol transport |
| ABCG5 (Q9H222) | CryoEM (2.60 Å, 7R89) | Liver, intestine | Heart disease, gallstone |

The ABC transporter protein names, their UniProt ID, and CryoEM structure in Å with PDB ID are listed. The lists of tissue location, medical relevance, and function are not exhaustive. Updated results become available from more and more recent studies.

ABCD1 is involved in the transport of various acetyl-CoAs from the cytosol to the peroxisomal lumen (Kawaguchi and Imanaka, 2022). ABCD1 is found in peroxisomes and plays a role in lipid homeostasis and is X-linked to adrenoleukodystrophy (Manor et al., 2021).

ABCD4 is found in lysosomes and participates in the transport of cobalamin (vitamin B12) from lysosomes into the cytosol. It is linked to methylmalonic aciduria and homocystinuria type cblJ (Hwang et al., 2021) and is also X-linked to adrenoleukodystrophy (Kawaguchi and Imanaka, 2022).

Both ABCG1 and ABCG5 share high protein sequence and structural similarity and are hemi-transporters that play a significant role in cholesterol transport and metabolism (Rezaei et al., 2022). ABCG1 seems to exclusively transport cholesterol as a homodimer found in macrophages, whereas the ABCG5/G8 heterodimer seems to transport both cholesterol and phytosterols and is found in the liver and the intestine. ABCG5 is involved in sterol trafficking and linked to sitosterolemia (Berge et al., 2000; Lee et al., 2001; Fong and Patel, 2021; Sun et al., 2021).

Recently, CryoEM has significantly transformed structural biology (Henderson et al., 2011; Vinothkumar and Henderson, 2016). With the new structural biology revolution, the molecular structures of several human ABC transporters have been experimentally determined including ABCB7 (Protein Data Bank [PDB]: 7VGF) (Yan et al., 2022), ABCC8 (PDB: 7S5V) (Zhao and Mackinnon, 2021), ABCD1 (PDB: 7SHM) (Wang et al., 2021), ABCD4 (PDB: 6JBJ) (Xu et al., 2019), ABCG1(PDB: 7R8D) (Rezaei et al., 2022), and ABCG5 (PDB: 7R89) (Berge et al., 2000; Lee et al., 2001; Fong and Patel, 2021; Sun et al., 2021). However, given that each ABC transporter typically has two transmembrane domains with 5–11 membrane spanning alpha helices, experimental structural determination requires detergent screening for structural stabilization before protein purification can be carried out. This process, however, is specific to each membrane protein and therefore is time-consuming and expensive (Vinothkumar and Henderson, 2010; Qing et al., 2022).

AlphaFold2, which was developed by Google DeepMind, is a revolutionary artificial intelligence tool that uses deep learning to predict protein structures from their amino acid sequences (Jumper et al., 2021; Varadi et al., 2021). DeepMind, in partnership with EMBL-EBI, released the AlphaFold Protein Structure Database which contains over 214 million predicted protein structures in July 2022 (Hassabis, 2022). In comparison to the ~213,000 experimentally determined protein structures available through RCSB-PDB, AlphaFold2's contribution is a significant advancement. However, AlphaFold2 predictions still have acknowledged limitations and need to be validated through experimental analysis.

Physical structural studies of the water-soluble QTY variants of membrane proteins are still needed to validate the AlphaFold2-predicted structures.

We previously applied the QTY (glutamine, threonine, tyrosine) code to design several detergent-free chemokine and cytokine receptors, all of which retained structural thermal stability and native ligand-binding activities despite substantial changes to the transmembrane domain (Zhang et al., 2018; Hao et al., 2020; Tegler et al., 2020). These water-soluble variants were then used to elucidate the mechanism of native receptor-ligand interaction and their binding abilities despite significant truncation in several chemokine receptors (Qing et al., 2019, 2020). Using the online version of AlphaFold2, we predicted the QTY variant structures of 7 chemokine receptors and 1 olfactory receptor (Skuhersky et al., 2021), 14 glucose transporters (Smorodina et al., 2022a), and 13 solute carrier transporters (Smorodina et al., 2022b). We recently also showed that the QTY code also works very well for bacterial outer membrane beta-barrels (Sajeev-Sheeja et al., 2023) and for IgG antibodies that are rich in beta-sheet structures (Li et al., 2023).

The AlphaFold2 online program provided results in ~1 hour or even minutes for each prediction, instead of weeks, as was the case before the advent of AlphaFold2. The predicted water-soluble variants share remarkable structural similarities with the native protein despite significant changes to the transmembrane domain and with decreases in hydrophobicity.

Here, we present bioinformatic studies of six experimentally determined human ABC transporters and their AlphaFold2-predicted water-soluble QTY variants. We provide superpositions of the hydrophobic native transporters and their hydrophilic QTY variants. We also provide the comparative hydrophobicity molecular structures with their hydrophilic QTY variants.

## Results and discussion

### *The rationale of the QTY code*

The QTY code is a simple code that systematically replaces four hydrophobic amino acids leucine (L), isoleucine (I), valine (V), and phenylalanine (F) with three neutral and polar amino acids glutamine (Q), threonine (T), and tyrosine (Y), respectively (T replaces both I and V), since the respective electron density maps between L *vs* Q, I & V *vs* T, and F *vs* Y show remarkable structural similarities (Zhang et al., 2018; Zhang and Egli, 2022). After applying the QTY code, the hydrophobic amino acids are replaced with hydrophilic ones, and the transmembrane domain loses its hydrophobic characteristic.

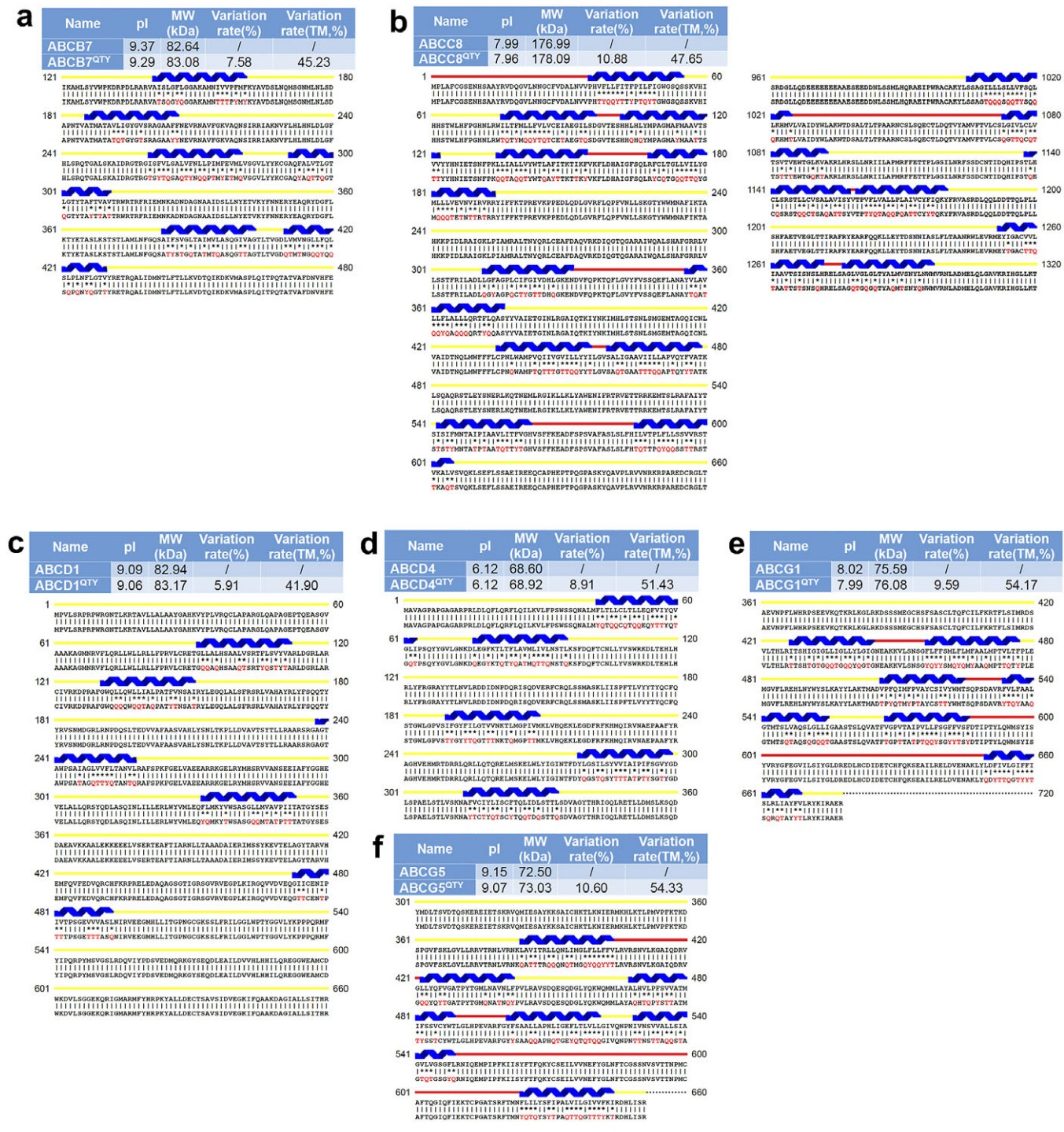

**Figure 1.** Protein sequence alignments of six native ABC transporters with their water-soluble QTY variants. The symbols | and * indicate whether amino acids are identical or different, respectively. Please note the Q, T, and Y amino acids (red) replacing L, V and I, and F, respectively. The alpha helices (blue) are shown above the protein sequences. The loop color codes are: internal (yellow) and external (red). The characteristics of natural and QTY variants listed are isoelectric (pI), molecular weight (MW), total variation %, and transmembrane variation %. The alignments are: a) ABCB7 vs ABCB7^QTY, b) ABCC8 vs ABCC8^QTY, c) ABCD1 vs ABCD1^QTY, d) ABCD4 vs ABCD4^QTY, e) ABCG1 vs ABCG1^QTY, and f) ABCG5 vs ABCG5^QTY. Although there are significant QTY changes in the TM alpha helices (41.90–54.33%), their changes in MW and pI are insignificant.

## ABC transporter protein sequence alignments and other characteristics

The protein sequences of the native ABC transporters and their QTY variants were aligned (Figure 1 and Figure S1 in the Supplementary Material for enlarged view). Despite overall changes to the amino acid composition (7.81–10.88%) and significant changes to the transmembrane domains (41.90–54.33%), the isoelectric-focusing points (pI) remain rather similar, with anywhere from a 0- to 0.08-unit difference. These pI changes are insignificant regarding surface charges and unlikely to disrupt delicate structures. Amino acids Q, T, and Y do not bear any charges at neutral pH and therefore do not significantly alter a protein's pI. Though the

molecular weights (MWs) of the QTY counterparts were slightly bigger than the native protein, this can be explained by the fact that the saturated carbon side chains of the native proteins are replaced by water-soluble -OH and -$NH_2$ in the QTY variants. Nitrogen (14 Da) and oxygen (16 Da) have high MW than carbon (12 Da).

### Superpositions of native CryoEM ABC transporters and their water-soluble QTY variants

We superposed the molecular structure of sixCryoEM native ABC transporters with their respective QTY variants. The structures of ABCB7 (PDB: 7VGF), ABCC8 (PDB: 7S5V), ABCD1 (PDB: 7SHM), ABCD4 (PDB: 6JBJ), ABCG1(PDB: 7R8D), and ABCG5 (PDB: 7R89) have already been experimentally determined via CryoEM, and so the superposition of the following protein pairs were carried out: ABCB7 *vs* ABCB7$^{QTY}$, ABCC8 *vs* ABCC8$^{QTY}$, ABCD1 *vs* ABCD1$^{QTY}$, ABCD4 *vs* ABCD4$^{QTY}$, ABCG1 *vs* ABCG1$^{QTY}$, and ABCG5 *vs* ABCG5$^{QTY}$. Thus, the CryoEM-determined structures and the AlphaFold2-predicted QTY variant structures were directly compared.

The native structures and the QTY variants superposed very well, with the exception of some unstructured loops and the N- and C-termini. For simplicity and clarity, these loops were removed in the figures. The root mean square deviation (RMSD) for the pairwise structures was between 1.064 and 3.413 Å, with most pairs having an RMSD of <2.00 Å (Table 2 and Figure 2). Figure 2 shows i) the CryoEM-determined native structures in magenta and the ii) AlphaFold2-predicted QTY variant structures in cyan of: a) ABCB7 *vs* ABCB7$^{QTY}$ (1.951 Å), b) ABCC8 *vs* ABCC8$^{QTY}$ (3.413 Å), c) ABCD1 *vs* ABCD1$^{QTY}$ (1.699 Å), d) ABCD4 *vs* ABCD4$^{QTY}$ (2.212 Å), e) ABCG1 *vs* ABCG1$^{QTY}$ (1.064 Å), and f) ABCG5 *vs* ABCG5$^{QTY}$ (1.637 Å). As seen in Figure 2, the structures share very similar folds despite a 41.90–54.33% replacement of amino acids in the transmembrane domain using the QTY code in the QTY variants. The results first confirm that AlphaFold2's predictions are highly accurate but also that the native ABC transporters and their water-soluble QTY variants share remarkable structural similarities.

### Superpositions of AlphaFold2-predicted native ABC transporters and their water-soluble QTY variants

Since the CryoEM structures were determined experimentally by different research groups, we asked how well would the structures superpose with the AlphaFold2-predicted native structures and their water-soluble QTY variants, as we did in our previous studies (Skuhersky et al., 2021; Smorodina et al., 2022a,b). We thus superposed the AlphaFold2-predicted native structures and their water-soluble QTY variants (Figure 3). As can be seen from Figure 3, these AlphaFold2-predicted structures superposed very well. For example: a) ABCB7 *vs* ABCB7$^{QTY}$ (RMSD = 0.913 Å), b) ABCC8 *vs* ABCC8$^{QTY}$ (RMSD = 1.409 Å), c) ABCD1 *vs* ABCD1$^{QTY}$ (RMSD = 1.290 Å), d) ABCD4 *vs* ABCD4$^{QTY}$ (RMSD = 1.383 Å), e) ABCG1 *vs* ABCG1 (RMSD = 0.387 Å), and f) ABCG5 *vs* ABCG5$^{QTY}$ (RMSD = 0.866 Å). These superpositions further indicate the water-soluble QTY variants share very similar structures with the native ABC transporters. Furthermore, we superposed CryoEM structures and the AlphaFold2-predicted native structures to validate AlphaFold3's capability and accuracy (Figure S2 in the Supplementary Material).

### Superpositions of CryoEM structures with AlphaFold2-predicted native ABC transporters and their water-soluble QTY variants

We also ask how well the superposition structures look like if we superpose i) the experimentally determined CryoEM ABC transporter structures with ii) AlphaFold2-predicted native transporters and iii) AlphaFold2-predicted water-soluble QTY variant transporters. These superpositions are shown in Figure 4. These three different kinds of structures apparently superpose very well. The difference and variations are insignificant (Figure 4). The three kinds of superpositions not only further validate the AlphaFold2 usefulness but also suggest that the water-soluble QTY variant ABC transporters could be expressed, purified, and used as soluble antigens to generate therapeutic monoclonal antibodies.

**Table 2.** The protein characteristics of six human ABC native transporters and their QTY variants

| Name | RMSD (Å) | pI | Mw (kDa) | TM variation (%) | Overall variation (%) |
|---|---|---|---|---|---|
| ABCB7$^{CryoEM}$ | – | 9.37 | 82.64 | – | – |
| ABCB7$^{QTY}$ | 1.951 Å | 9.29 | 83.08 | 45.23 | 7.58 |
| ABCC8$^{CryoEM}$ | – | 7.99 | 176.99 | – | – |
| ABCC8$^{QTY}$ | 3.413 Å | 7.96 | 178.09 | 47.65 | 10.88 |
| ABCD1$^{CryoEM}$ | – | 9.09 | 82.94 | – | – |
| ABCD1$^{QTY}$ | 1.699 Å | 9.06 | 83.17 | 41.90 | 5.91 |
| ABCD4$^{CryoEM}$ | – | 6.12 | 68.60 | – | – |
| ABCD4$^{QTY}$ | 2.212 Å | 6.12 | 68.92 | 51.43 | 8.91 |
| ABCG1$^{CryoEM}$ | – | 8.02 | 75.59 | – | – |
| ABCG1$^{QTY}$ | 1.064 Å | 7.99 | 76.08 | 54.17 | 9.59 |
| ABCG5$^{CryoEM}$ | – | 9.15 | 72.50 | – | – |
| ABCG5$^{QTY}$ | 1.637 Å | 9.07 | 73.03 | 54.33 | 10.60 |

Abbreviations: pI, isoelectric focusing; MW, molecular weight, TM, transmembrane, (−), not applicable, and RMSD, residue mean square distance.
The six ABC transporters are listed in the same order as the Figure 1. RMSDs were calculated after missing residuals (unstructured loops) in the native CryoEM-determined structures and the corresponding residuals in the predicted QTY structures were cut out. If the native protein was a dimer, one monomer was also cut out. For ABCG5, the CryoEM-determined structure was a heterodimer of ABCG5/G8. ABCG8 was removed for clarity. The QTY amino acid substitutions in the transmembrane (TM) are significant between 41.90% and 54.33%, whereas the overall structural changes are between 5.91% and 10.88%.

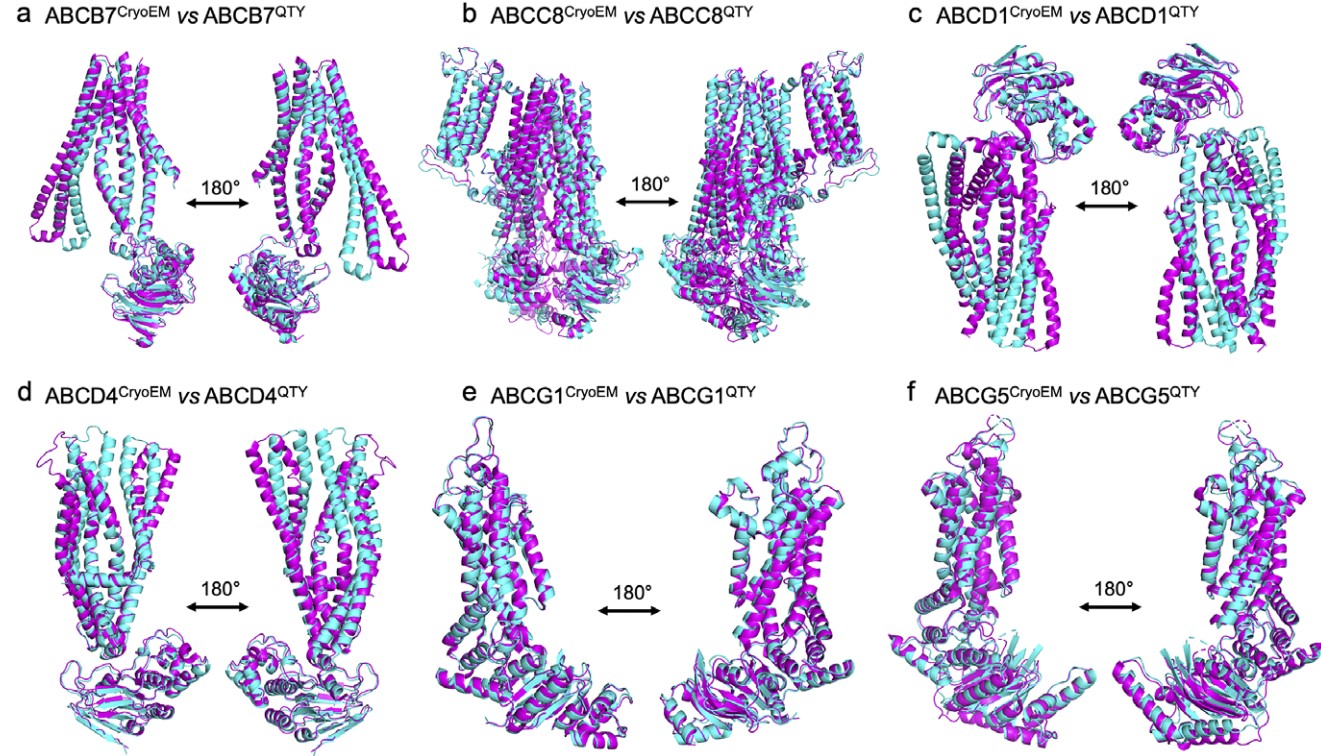

**Figure 2.** Superpositions of six human Cryo-EM-determined structures of native ABC transporters and their AlphaFold2-predicted water-soluble QTY variants. The CryoEM-determined structures of the native transporters are obtained from the Protein Data Bank (PDB). The CryoEM structures (magenta) are superposed with their QTY variants (cyan) predicted by AlphaFold2. These superposed structures show that the native transporters and their QTY variants have very similar structures. For clarity of direct comparisons, unstructured loops in the CryoEM structures were removed in the QTY variants. Similarly, one of the monomers in homodimers and ABCG8 in the ABCG5/G8 heterodimer was cut out for clarity. a) ABCB7 *vs* ABCB7$^{QTY}$, b) ABCC8 *vs* ABCC8$^{QTY}$, c) ABCD1 *vs* ABCD1$^{QTY}$, d) ABCD4 *vs* ABCD4$^{QTY}$, e) ABCG1 *vs* ABCG1$^{QTY}$, and f) ABCG5 *vs* ABCG5$^{QTY}$.

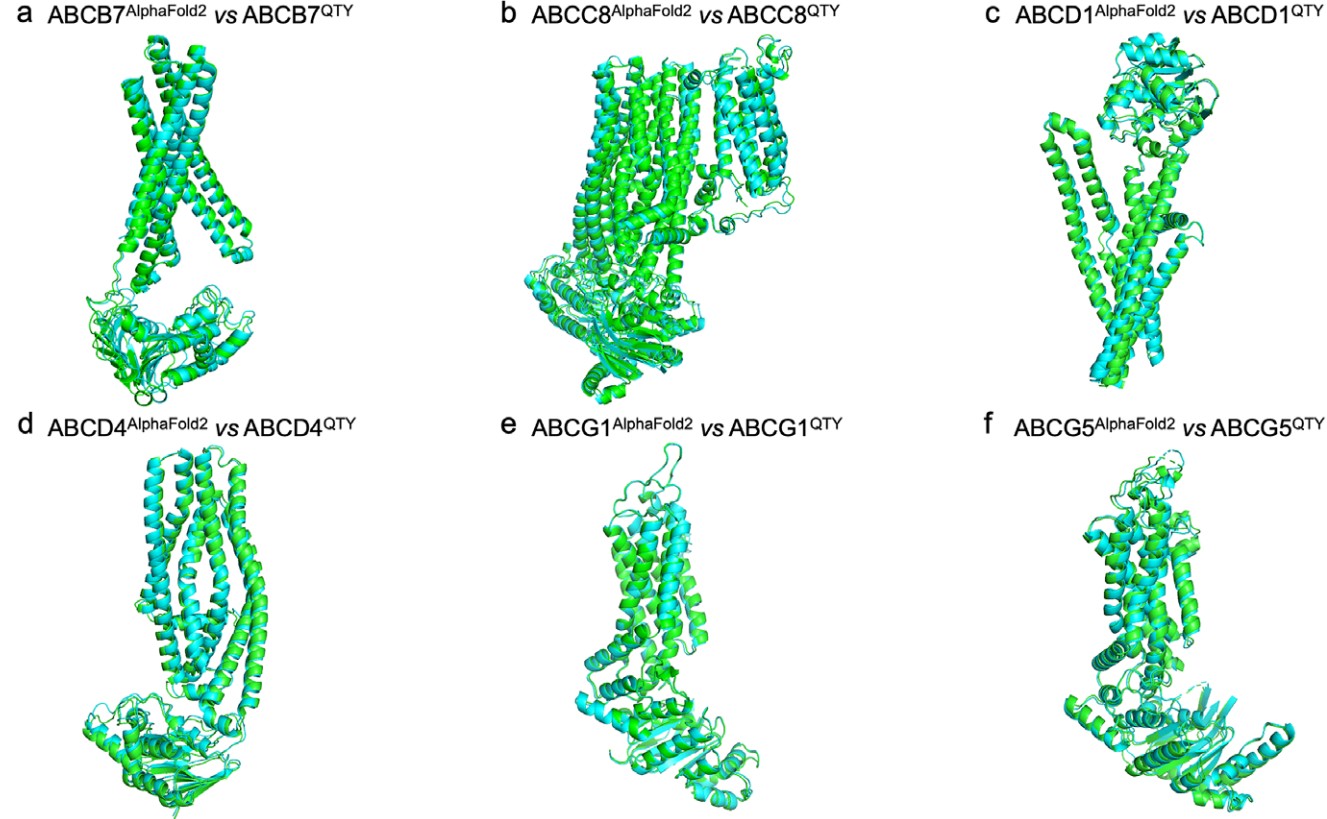

**Figure 3.** Superpositions of AlphaFold2-predicted structures of native and their QTY variants. Color code: green = AlphaFold2-predicted native structures; cyan = AlphaFold2-predicted water-soluble QTY variants. a) ABCB7 *vs* ABCB7$^{QTY}$ (RMSD = 0.913 Å), b) ABCC8 *vs* ABCC8$^{QTY}$ (RMSD = 1.409 Å), c) ABCD1 *vs* ABCD1$^{QTY}$ (RMSD = 1.290 Å), d) ABCD4 *vs* ABCD4$^{QTY}$ (RMSD = 1.383 Å), e) ABCG1 *vs* ABCG1 (RMSD = 0.387 Å), and f) ABCG5 *vs* ABCG5$^{QTY}$ (RMSD = 0.866 Å).

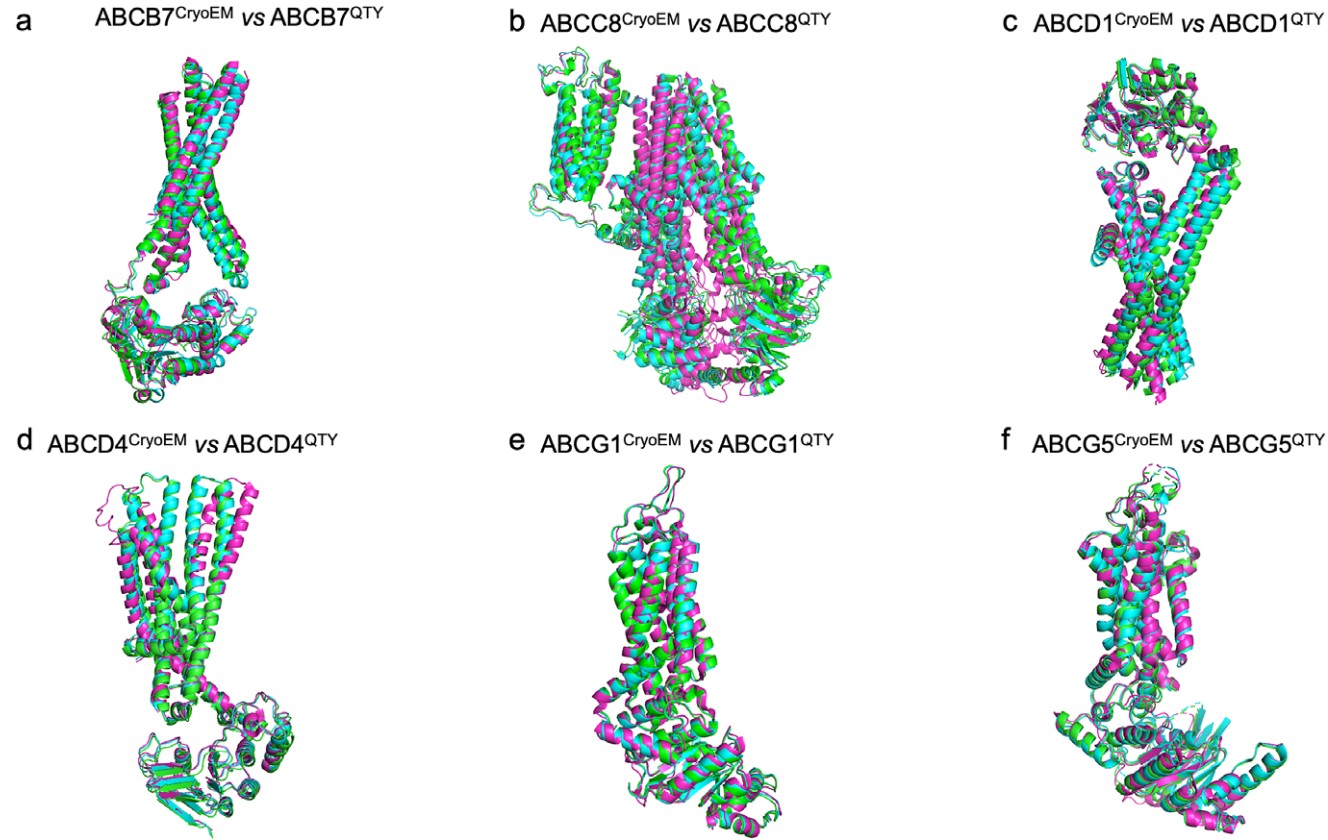

**Figure 4.** Superpositions of CryoEM structures with AlphaFold2-predicted native ABC transporters and their water-soluble QTY variants. Superposition of **i**) the experimentally determined CryoEM ABC transporter structures (magenta) with **ii**) AlphaFold2-predicted native transporters (green) and **iii**) AlphaFold2-predicted water-soluble QTY variant transporters (cyan). These superpositions are shown in Figure 4. These three different kinds of structures are apparently superposed very well. The difference and variations are insignificant. These three kinds of superpositions not only further validate the AlphaFold2 usefulness but also show that the water-soluble QTY variant ABC transporters could be used as soluble antigens to generate therapeutic monoclonal antibodies. a) ABCB7$^{CryoEM}$/ABCB7$^{Native}$/ABCB7$^{QTY}$, b) ABCC8$^{CryoEM}$/ABCC8$^{Native}$/ABCC8$^{QTY}$, c) ABCD1$^{CryoEM}$/ABCD1$^{Native}$/ABCD1$^{QTY}$, d) ABCD4 $^{CryoEM}$/ABCD4$^{Native}$/ABCD4$^{QTY}$, e) ABCG1$^{CryoEM}$/ABCG1$^{Native}$/ABCG1$^{QTY}$, f) ABCG5$^{CryoEM}$/ABCG5$^{Native}$/ABCG5$^{QTY}$.

We also superposed the transmembrane domains only, by cutting out everything that is not a part of the transmembrane domains. The transmembrane domains are adjusted after amino acids in the nontransmembrane domain are removed. Here, we present the superposed CryoEM structures and their AlphaFold2-predicted water-soluble QTY variants, with reasonable RMSD values (Figure S3 in the Supplementary Material). The RMSD values are: a) ABCB7$^{CryoEM}$ *vs* ABCB7$^{QTY}$ (RMSD = 0.686 Å), b) ABCC8$^{CryoEM}$ *vs* ABCC8$^{QTY}$ (RMSD = 1.390 Å), c) ABCD1$^{CryoEM}$ *vs* ABCD1$^{QTY}$ (RMSD = 2.338 Å), d) ABCD4$^{CryoEM}$ *vs* ABCD4$^{QTY}$ (RMSD = 3.787 Å), e) ABCG1$^{CryoEM}$ *vs* ABCG1$^{QTY}$ (RMSD = 0.539 Å, and f) ABCG5/G8$^{CryoEM}$ *vs* ABCG5/G8$^{QTY}$ (RMSD = 0.701 Å).

### Analysis of the hydrophobic surface of native ABC transporters and the water-soluble QTY variants

The native ABC transporters are highly hydrophobic, especially in the 5–11 transmembrane alpha helical domains. Thus, the native proteins require detergents to solubilize after being removed from the lipid bilayer membranes. Without the appropriate detergents, they immediately aggregate, precipitate, and lose their biological functions.

The 5–11 transmembrane alpha helices are directly embedded in the hydrophobic lipid bilayer, so the hydrophobic side chains of amino acids leucine (L), isoleucine (I), valine (V), and phenylalanine (F) directly interact with the lipid molecules and exclude water. Thus, the transmembrane domains exhibit highly hydrophobic patches (Figure 5).

After systematic replacement of hydrophobic amino acids L, I, V, F, with hydrophilic amino acids glutamine (Q), threonine (T), and tyrosine (Y), using the QTY code, these hydrophobic patches are largely reduced (Figure 5). Transforming the transmembrane alpha helices from hydrophobic to hydrophilic using the QTY code did not significantly alter the alpha helix structures. This would have been a rather unexpected result; however, our previous biochemistry experiments demonstrated that the QTY variants of chemokine and cytokine receptors retained structural integrity, stability, and ligand-binding activity even after becoming water soluble.

### AlphaFold2 predictions

For over six decades, structural biologists and protein scientists have sought to predict how proteins fold naturally and seemingly instantly posttranslation. AlphaFold2's accurate predictions now enable us to study protein structure in more detail *in silico* and obtain previously unattainable protein structures, especially integral transmembrane proteins, at least in the framework.

We can now approach transmembrane structures computationally first by using AlphaFold2 predictions to compare native protein structures with AlphaFold2-predicted water-soluble QTY variants.

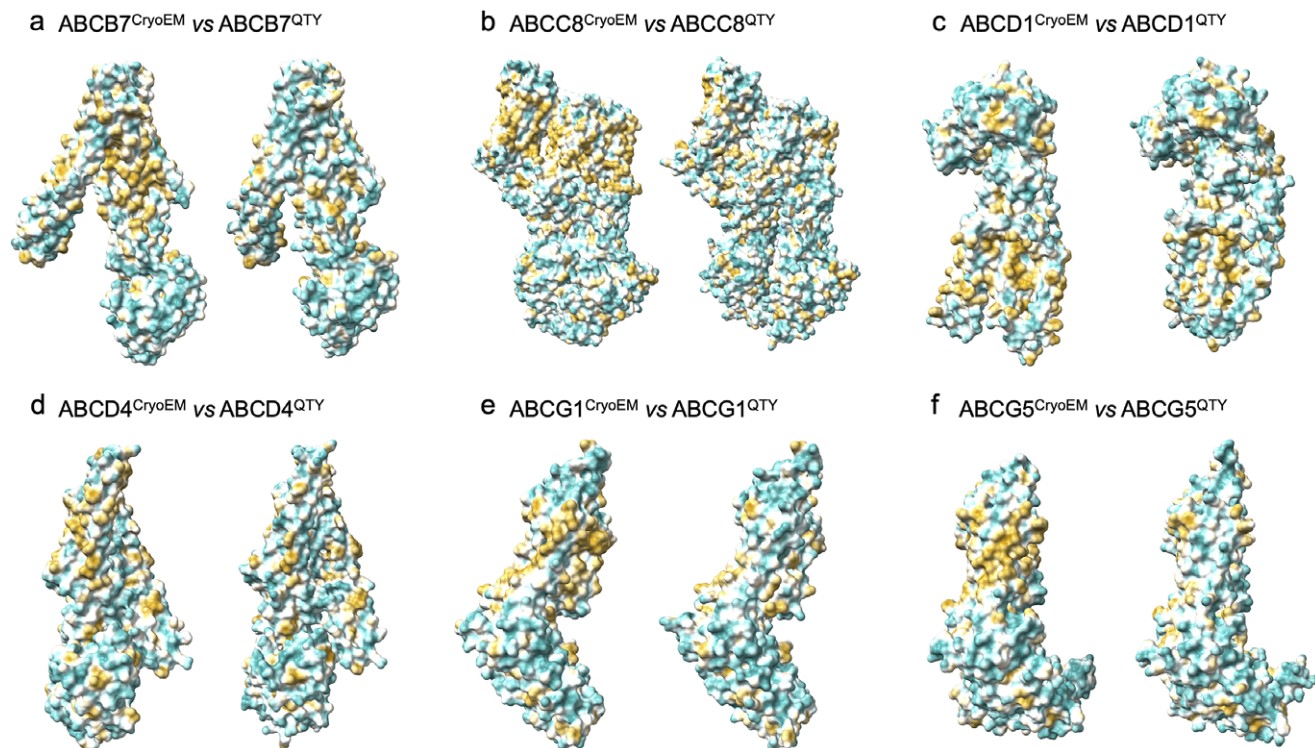

**Figure 5.** Hydrophobic surface of six native ABC transporters and their water-soluble QTY variants. The native ABC transporters have many hydrophobic residues L, I, V, and F in the transmembrane helices. After Q, T, and Y substitutions of L, I and V, and F respectively, the hydrophobic surface patches (yellowish) in the transmembrane helices become more hydrophilic (cyan). For clarity of direct comparisons, unstructured loops in the CryoEM structures were removed in the QTY variants. Similarly, one of the monomers in homodimers and ABCG8 in the ABCG5/G8 heterodimer was cut out for clarity. a) ABCB7 *vs* ABCB7[QTY], b) ABCC8 *vs* ABCC8[QTY], c) ABCD1 *vs* ABCD1[QTY], d) ABCD4 *vs* ABCD4[QTY], e) ABCG1 *vs* ABCG1[QTY], and f) ABCG5 *vs* ABCG5[QTY].

We can then express and purify the water-soluble QTY variants. Although AlphaFold2 predicts less accurate flexible loops, there are some deviations between the predicted protein structure and the real protein's structure. This is expected as proteins are dynamic entities, changing their conformation and structure when performing functions such as enzyme catalysis, ligand bindings, and molecular transport.

In collaboration with the European Bioinformatics Institute (EBI), DeepMind has made over 214 million predicted protein structures available online (https://alphafold.ebi.ac.uk). This number will continuously increase over time.

We previously used AlphaFold2 to predict the QTY variants of chemokine receptors (Skuhersky et al., 2021), glucose transporters (Smorodina et al., 2022a), and solute carrier transporters (Smorodina et al., 2022b) and directly superposed them to experimentally determined molecular structures. The speed and accuracy of AlphaFold2 in predicting our QTY-designed membrane proteins is unprecedented. Instead of taking weeks to predict a single structure, AlphaFold2 can predict a new structure in ~1 hour, or even minutes for smaller proteins. AlphaFold2 not only significantly accelerates studies of protein structures but also the designs of new proteins, the discovery of new protein interactions, and perhaps previously unknown or new functions.

### The rationale for selecting ABC transporters for this study

In our initial QTY code for protein design studies, we first selected one chemokine receptor CXCR4, which is a G protein-coupled receptor family with seven transmembrane alpha helices, for the QTY code design. We systematically carried out biochemistry, biophysics, and molecular biology experiments and obtained reproducible ligand binding results and protein thermostability results (Zhang et al., 2018). The surprising results were viewed with skepticism. We then added another chemokine receptor CCR5. We again obtained reproducible experimental ligand binding results. However, in order to further convince ourselves and others, we selected a total of seven chemokine receptors and four cytokine receptors to carry out a wide range of biochemistry, biophysics, and molecular biology experiments. The QTY code worked very well for these 11 receptors (Zhang et al., 2018; Qing et al., 2019; Hao et al., 2020). We then asked if the QTY code also works for structural bioinformatic studies for glucose transporters that have 14 transmembrane alpha helices (Smorodina et al., 2022a) and solute carrier transporters with 12 transmembrane alpha helices (Smorodina et al., 2022b). They also worked well. We then asked if QTY code also works for ABC transporters and membrane proteins.

ABC transporters belong to one of the largest transporter family and possibly one of the oldest transporter gene families that involves in many important biological activities. We thus are interested to carry out the structural bioinformatic study of the six representative ABC transporters with known CryoEM structures to further validate the QTY code. Some ABC transporters in human are involved in tumor drug resistance, multidrug resistance, and genetic diseases including cystic fibrosis, X-linked adrenoleukodystrophy, Stargardt disease, Dubin–Johnson syndrome, Byler's disease, ataxia, and more (Engelen et al., 2014, Robey et al., 2018; Dean et al., 2022; Kawaguchi and Imanaka, 2022).

The water-soluble QTY variants of the ABC transporters may be very useful: i) use the water-soluble ABC transporter as antigens to generate therapeutic monoclonal antibodies for the treatment of various diseases, ii) use the purified soluble transporter proteins to carry out high-throughput drug discoveries, and iii) to facilitate to determine the molecular structures of other ABC transporters that currently have no experimentally determined structures.

## Conclusion

There are three distinct types of alpha helices in nature: i) the hydrophilic alpha helices including those found in hemoglobin, lysozyme, and many other water-soluble enzymes and circulating proteins such as growth factors, cytokines, and antibodies, ii) the hydrophobic alpha helices including those found in integral transmembrane proteins found in G protein-coupled receptors, transporters, various ion channels, and photosynthesis systems, and iii) the amphiphilic alpha helices with both hydrophilic and hydrophobic amino acid residues. These three types of alpha helices have nearly identical molecular structures despite differences in hydrophobicity and hydrophilicity (Pauling et al., 1951; Branden and Tooze, 1999; Fersht, 2017; Liljas et al., 2017; Zhang and Egli, 2022). This insight is the structural and molecular basis of the QTY code.

Applying the QTY code, our study presents a straightforward approach to systematically convert hydrophobic alpha helices to hydrophilic alpha helices in ABC transporters, rendering them into water-soluble variants. We structurally and bioinformatically analyzed the sequences and structures of six human CryoEM ABC transporters and their six QTY variants. The structures of QTY variants showed a global similarity to native proteins, suggesting that the QTY variant proteins are likely to retain their functions. To validate this assumption, we employed various in silico computational and bioinformatic tools to calculate sequence and structure characteristics associated with protein stability and water solubility. These structures of the native ABC transporters and their water-soluble QTY variants are highly similar despite significant changes to the amino acid sequence in transmembrane domains and significant reduction in hydrophobic surfaces, thus demonstrating that the QTY code is a viable approach to modeling water-soluble variants of integral membrane proteins. We believe that the hydrophilic ABC transporters hold potential for applications as water-soluble antigens for the discovery of therapeutic monoclonal antibodies for use in the treatment of human diseases including cancers.

## Methods

### Protein sequence alignments and other characteristics

The native protein sequences for ABC transporters including ABCB7, ABCC8, ABCD1, ABCD4, ABCG1, and ABCG5 were obtained from UniProt (https://www.uniprot.org). The sequences for the QTY variants were aligned using the same methods as previously described. The MWs and pI values of the proteins were calculated using the Expasy (https://web.expasy.org/compute_pi/).

### AlphaFold2 predictions

The protein structures of the QTY variants were predicted using AlphaFold2 (https://github.com/sokrypton/ColabFold) and by following the instructions at the website. PBD files for the predicted native protein structures were obtained from The EBI (https://

alphafold.ebi.ac.uk), which contains all AlphaFold2-predicted structures for native proteins. The UniProt website (https://www.uniprot.org) provided protein ID, entry name, description, and FASTA sequence for each native protein. The QTY code can be applied to FASTA sequences by manually replacing amino acids in the TM domains (found on the Protter 2D diagram plotting website, http://wlab.ethz.ch/protter/start/) but can also be done through the QTY method website (https://pss.sjtu.edu.cn/). The website also provides MWs, pI values, TM variation, and overall variation.

### Superposed structures

PBD files for native protein structures experimentally determined by CryoEM were taken from the PDB include ABC transporters ABCB7 (PDB: 7VGF), ABCC8 (PDB: 7S5V), ABCD1 (PDB: 7SHM), ABCD4 (PDB: 6JBJ), ABCG1(PDB: 7R8D), and ABCG5 (PDB: 7R89). Predictions for the QTY variants were carried out using the AlphaFold2 program, which can be found at https://github.com/sokrypton/ColabFold. These structures were superposed and the RMSDs were calculated using PyMOL (https://pymol.org/2/). For ABCB7, ABCD1, ABCD4, and ABCG1, the CryoEM molecular structure models the homodimers of the proteins, and for ABCG5, the CryoEM molecular structure models the heterodimer of ABCG5 and ABCG8. On the other hand, the AlphaFold2-predicted QTY variants only models the monomer. For simplicity and clarity, unstructured loops and one of the protein monomers were manually removed from the figures.

### Structure visualization

PyMOL (https://pymol.org/2/) was used to superpose the native protein structure and the QTY variant. UCSF Chimera (https://www.rbvi.ucsf.edu/chimera) was used to render each protein model with hydrophobicity patches.

**Data availability of AlphaFold2-predicted water-soluble QTY variants**

European Bioinformatics Institute (EBI, https://alphafold.ebi.ac.uk) serves as a database for more than 214 million AlphaFold2-predicted protein structures. Please use the website for more detailed information, and please contact the first author Emily Pan at emilypan2006@gmail.com. Protein characteristics used in the analysis are available on UniProt (https://www.uniprot.org/). The native CryoEM-determined ABC transporter proteins are available in the RCSB PDB repository (https://www.rcsb.org/). The QTY code designed water-soluble variants of the proteins are available at https://github.com/emilypan2024/abc-transporters.

**Open peer review.** To view the open peer review materials for this article, please visit http://doi.org/10.1017/qrd.2024.2.

**Supplementary material.** The supplementary material for this article can be found at https://doi.org/10.1017/qrd.2024.2.

**Data availability statement.** The AlphaFold2-predicted protein structures are at European Bioinformatics Institute (EBI; https://alphafold.ebi.ac.uk). The QTY code designed water-soluble variants are published in this article, and protein characteristics used in the analysis are available on UniProt (https://www.uniprot.org/). The native CryoEM-determined ABC transporter proteins are available in the RCSB PDB repository (https://www.rcsb.org/). The QTY code designed water-soluble variants of the proteins are available at https://github.com/emilypan2024/abc-transporters.

**Acknowledgements.** We thank Dorrie Langsley for English editing.

**Author contribution.** Conceptualization: S.Z.; Data curation: E.P.; Formal analysis: E.P.; Investigation: E.P.; Methodology: E.S., F.T.; Review and editing: E.P., E.S., F.T., S.Z; Validation: E.P., E.S.; Writing – original draft preparation: E.P., S.Z..

**Financial support.** E.P. is a high school student, and E.S. is an undergraduate student. There are no funders who select the subject, conducting research, writing the manuscript, and where to publish.

**Competing interest.** Massachusetts Institute of Technology (MIT) filed several patent applications for the QTY code for GPCRs and $OH_2$ Laboratories licensed the technology from MIT to work on water-soluble GPCR variants. However, this article does not study GPCRs. S.Z. is an inventor of the QTY code and has a minor equity of $OH_2$ Laboratories. S.Z. founded a startup 511 Therapeutics to generate therapeutic monoclonal antibodies against solute carrier transporters to treat pancreatic cancer. S.Z. has majority equity in 511 Therapeutics. This article does not study human ABC transporters. S.Z. currently takes an unpaid leave from MIT. All other authors have no competing interest.

**Ethics statement.** All methods were carried out in accordance with relevant guidelines and regulations. All experimental protocols were approved by a named institutional and licensing committee. Neither human biological samples nor human subjects were used in the study. This is a completely digital structural bioinformatic study using the publicly available AlphaFold2 machine learning program.

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
