## [Reviewer Report]

This manuscript present the bioinformatic study of six QTY designed ABC transporters and compare them in several ways to native proteins. It shows a nice addition to other classes of QTY designed membrane proteins. Yet there are a few issues the author should address before the manuscript can be accepted.

1. The authors have reported the QTY designs of other types of transporters. Can the author elaborate if the type of the transporters have any impact on the successful QTY design of the proteins?

2. Can the author provide more discussion on the potential use of QTY designs of these proteins?

3. What are the RMSDs for only the transmembrane regions for these proteins?

4. It would be nice if the AF2 predicted structure of native proteins can be added to superimposition.

5. The grammar needs to be carefully checked, such as in the abstract: “often results in cancer patient treatment poor outcome.”

---

## [Reviewer Report]

The manuscript by Pan et al. is a continuation of the studies of replacement of hydrophobic residues in proteins by the QTY-code. In this case the objects of the study are membrane bound ABC transporters that can be involved in multi-drug resistance. The aim is to generate soluble proteins that can be used to produce monoclonal antibodies that can be used clinically against drug resistance.

The study uses six ABC transporters the structures of which were already determined by cryo-EM and published and compared with the structures of the same proteins where the hydrophobic residues of the transmembrane helices were replaced by the QTY method. First the resolution in the structure determination is moderate (2.6-3.6Å). This makes the accuracy of the atomic coordinates limited. In the comparison of atomic coordinates, the resolution of the structures plays a significant role. It would be valuable to also compare the structures determined by the same methods, in this case AlphaFold2.

In the replacement of hydrophobic residues in the helices no respect has been taken whether the residues are situated on the surface or are in the inside of the protein. A hydrophilic residue (QTY) in the interior of a protein is always in need to find a matching hydrogen bond donor or acceptor. In the interior of proteins, one is unlikely to find hydrogen bonding partner and will cause a local disturbance of the structure.

To generate a soluble ABC transporter with limited structural disturbances one should use the available structures to modify only the hydrophobic residues on the outside, maybe not only on the helices and keep the interior residues unmodified. In addition, it would be valuable to compare the native protein structures determined by AlphaFold2 compared with the modified ones. My suggestion is that a manuscript with these modifications could be published in QRB.